# Benefits of Participation in Clinical Trials: An Umbrella Review

**DOI:** 10.3390/ijerph192215368

**Published:** 2022-11-21

**Authors:** Amira Bouzalmate-Hajjaj, Paloma Massó Guijarro, Khalid Saeed Khan, Aurora Bueno-Cavanillas, Naomi Cano-Ibáñez

**Affiliations:** 1Department of Preventive Medicine and Public Health, Faculty of Medicine, University of Granada, 18016 Granada, Spain; 2Preventive Medicine Unit, Universitary Hospital Virgen de las Nieves, 18014 Granada, Spain; 3Instituto de Investigación Biosanitaria de Granada (IBS.GRANADA), 18012 Granada, Spain; 4CIBER de Epidemiología y Salud Pública (CIBERESP-Spain), 28029 Madrid, Spain

**Keywords:** participation, non-participants, systematic reviews, umbrella review, health changes, randomised controlled trials

## Abstract

Participation in randomised clinical trials (RCTs) entails taking part in the discovery of effects of health care interventions. The question of whether participants’ outcomes are different to those of non-participants remains controversial. This umbrella review was aimed at assessing whether there are health benefits of participation in RCTs, compared to non-participation. After prospective registration (PROSPERO CRD42021287812), we searched the Medline, Scopus, Web of Science and Cochrane Library databases from inception to June 2022 to identify relevant systematic reviews with or without meta-analyses. Data extraction and study quality assessment (AMSTAR-2) were performed by two independent reviewers. Of 914 records, six systematic reviews summarising 380 comparisons of RCT participants with non-participants met the inclusion criteria. In two reviews, the majority of comparisons were in favour of participation in RCTs. Of the total of comparisons, 69 (18.7%) were in favour of participation, reporting statistically significant better outcomes for patients treated within RCTs, 264 (71.7%) comparisons were not statistically significant, and 35 (9.5%) comparisons were in favour of non-participation. None of the reviews found a harmful effect of participation in RCTs. Our findings suggest that taking part in RCTs may be beneficial compared to non-participation.

## 1. Introduction

Patients participating in randomised clinical trials (RCT) take part in discovering the effects of healthcare interventions. Eligible participants enrol in RCTs voluntarily in the hope that, in addition to the possibility to obtain a health improvement individually, their participation will benefit health status in future patients. In fact, given that RCT implementation requires approval by an ethics committee, requires oversight with regard to compliance with the protocol, and involves the support of extra research staff in the monitoring of care, it is likely that this surveillance and additional healthcare might result in the accrual of benefits compared to the usual care provided, regardless of the study arm allocation [1]. However, whether their outcomes are different to those of non-participants remains controversial [2,3,4,5,6].

Informed consent forms offered to patients before their enrolment into RCTs provide information about potential benefits and risks [7], but not those of participation *per se*, even for the control group. The successful recruitment of patients relies on active and personalised strategies [8] and depends on the confidence of patients and health professionals regarding the benefits and safety of RCTs. A recent review showed that the decision to participate in a surgical trial is influenced by patients’ abilities to make sense of the trial and trial processes, to weigh the risks and benefits of the treatment options, and to trust in the RCT staff [9]. In a meta-analysis of barriers to cancer clinical trial participation, physician and patient decision-making was identified as the reason for not enrolling by one out of four patients, beyond trial availability or clinical ineligibility [10]. In a cross-sectional study on attitudinal discordance between cancer patients and clinicians/research providers regarding RCT participation, patients more frequently reported negative beliefs, such as the belief that participation did not help patients personally (32.9% vs. 1.8%, *p* < 0.001), although they were more confident regarding the benefit risk ratio (57% vs. 44%, *p* = 0.03) and less concerned about treatment toxicity (18% vs. 60% *p* = 0.006) and randomisation or receiving a placebo (27% vs. 71% *p* = 0.005) [11]. In a qualitative study on participation in oncological therapy RCTs, health professionals reported that misconceptions based on negative beliefs and attitudes towards research were the main patient-level barriers to participation [12]. In a review, uncertainty about the risk-benefit ratio of clinical trial participation may lead to a magnification in the perceived likelihood to suffer an adverse event and reduce patients’ predisposition to participate, as well as making clinicians, especially oncologists, reluctant to offer their patients the opportunity to enrol in a clinical trial, so as not to jeopardise their therapeutic long-term relationship [13].

A patient and public involvement (PPI) approach to the trial development process, from the formulation of research questions to the dissemination of results, may help staff build trusting relationships with potential participants and foster mutual commitment [14,15]. If it can be demonstrated that participating in RCTs improves health status, this would encourage volunteers to take part in research and enable health professionals to be confident about inviting patients to engage in trials [16,17]. Evidence regarding the benefits of participating in RCTs may help to interpret the generalisability of research findings, aiding in the implementation of new interventions in clinical practice and healthcare policy [18]. In this umbrella review, we aimed to determine if there was a health benefit (outcome) among eligible people (population) from participation in RCTs (intervention) vs. non-participation (comparison group).

## 2. Material and Methods

We performed this umbrella review after prospective registration (PROSPERO number: CRD42021287812) and reported it in accordance with the relevant guidelines [19,20]. We also adhered to the reporting guidelines for overviews of reviews of healthcare interventions (PRIOR) [21].

### 2.1. Literature Search and Selection

We conducted a sensitive literature search without language restrictions in electronic databases (the Medline, Scopus, Web of Science and Cochrane libraries) from inception to June 2022. We used a combination of keywords and terms including “participation”, “non-participants”, “systematic reviews”, “meta-analysis”, “health changes”, “health status improvement”, “harmful”, and “randomized controlled trials”. All citations found were exported to Endnote software, where duplicates were removed. Two reviewers (ABH and PMG) carried out the search strategy independently using electronic databases and manual searches, and screened all abstracts and titles (Appendix A). 

We included studies aimed at assessing benefits or hazards of participation in RCTs independently of the intervention or control group allocation, compared to similar non-participating patients receiving conventional care outside of trials. The exclusion criteria were: studies which did not report benefits or harmful effects in all participants; study designs other than systematic reviews or meta-analyses on RCT, i.e., narrative reviews and reviews on non-RCTs; and reviews on effectiveness comparing intervention groups *versus* control groups, without comparisons with those outside the RCT. Any disagreement regarding the inclusion of the citations was resolved by obtaining the opinion of a third researcher (NCI). We contacted authors to obtain full-text articles that were not available. Finally, the selection of records was based on an independent review of the full texts to ensure that the inclusion and exclusion criteria had been fulfilled.

### 2.2. Data Extraction and Risk-of-Bias Assessment

The characteristics of selected studies were extracted independently by two reviewers (ABH and PMG) after reading the full text. We used a predefined form for data extraction, including citation details (author and year); objectives; characteristics and number of participants; the number of databases sourced and searched; the date range of the database search; the publication date range of studies included in the review that informed each outcome of interest; the instrument used to appraise the primary studies; and the ratings of their quality, comparator, type of intervention, and outcomes reported that were relevant to the umbrella review question.

The quality of the included systematic reviews was independently assessed by two reviewers (ABH and NCI). We chose the 16-item questionnaire “A Measurement Tool to Assess Systematic Reviews” (AMSTAR-2) [22,23] because of its more extensive use in umbrella reviews to assess quality, compared with other tools [24]. Disagreements were resolved via consultation with a third reviewer (PMG). According to the guidelines, the reviewers assigned one of four global quality ratings (i.e., high, moderate, low or critically low) after the consideration of 16 potential critical and noncritical weaknesses. High and moderate ratings reflected the presence of one or less or one noncritical weakness, respectively, whereas low and critically low ratings indicated one or more than one critical weakness, respectively.

### 2.3. Data Synthesis

The extracted data in each review were structured according to the PICO framework, noting the participant characteristics, intervention, comparator and outcome of each study. The findings were tabulated, including the overall number of RCTs and participants, the number of studies in favour and not in favour of participation [25] and whether meta-analysis and heterogeneity assessments were performed.

We also calculated the corrected covered area (CCA), a validated method of quantifying the degree of overlap between two or more reviews to help the decision process. CCA is expressed as a percentage, and is calculated as *(N − r)/(rc − r),* where *N* is the number of publications included in the evidence synthesis, *r* is the number of rows and *c* is the number of columns. Overlap is categorised as very high (CCA >15%), high (CCA 11–15%), moderate (CCA 6–10%) or slight (CCA 0–5%) [26]. In overlapping cases, we planned to give preference to the most recent review that had the highest quality (AMSTAR-2 assessment), supplied pooled-effect estimates or conducted a meta-analysis and had the highest number of studies or participants [27].

## 3. Results

### 3.1. Selection, Characteristics and Quality of Studies

A total of 914 records were initially identified. Six articles met the eligibility criteria (292 RCTs, 380 unique comparisons). The dates used for the searching of the databases ranged from 1880 [28] to 2017 [29]. Figure 1 displays a PRISMA flow diagram of the selection process. We have also provided a list of studies that might appear to meet the inclusion criteria but were excluded, with the main reason for their exclusion (Appendix A). The main characteristics of the selected reviews and meta-analyses are summarised in Table 1. 

Evidence maps of effect direction, association strength, evidence certainty of RCT participation and its benefits, heterogeneity and whether meta-analysis was performed are provided in (Appendix A). Among the six reviews included in this study, three performed meta-analyses [28,29,30]. Primary original research studies included in the reviews were conducted in different medical areas and included a wide range of interventions, such as medical, surgical or counselling interventions. A moderate degree of overlap was found (CCA = 10%) (Appendix A). The quality was high in three reviews [29,30,31], low in one [28] and critically low in two (Figure 2). Among the AMSTAR-2 criteria, the most frequent critical weaknesses identified were the lack of a comprehensive literature search strategy (50%) and an inappropriate investigation of publication bias (65%).

### 3.2. Synthesis of Findings

In two of our reviews [31,32] the majority (>50%) of comparisons were in favour of participation in RCTs. Of the total number of comparisons included, 69 (18.7%) were in favour of participation, reporting statistically significant better outcomes for patients treated within RCTs, and 264 (71.7%) comparisons were not statistically significant, whereas 35 (9.5%) comparisons were in favour of non-participation (Figure 3). None of the reviews showed a harmful effect of participation in RCTs in their overall synthesis.

### 3.3. Findings from the High-Quality Subgroup of Reviews

In a cancer review [31], 27/49 (55.1%) comparisons reported statistically significantly better outcomes in RCT participants, 12/49 (24.5%) comparisons were in favour of non-participation and 10/49 (20.4%) comparisons were non-significant. A meta-analysis comparing women’s health outcomes in obstetrics and gynaecology trials [29] found 3/21 (14.2%) comparisons in favour of participation, 1/21 (4.8%) comparisons in favour of non-participation and 17/21 (81%) non-significant comparisons. In another review regarding general medicine [30], a total of 11/141 (7.8%) comparisons were in favour of participation, reporting statistically significant better outcomes, lower complications and relapse for patients treated within RCTs, whereas 10/141 (7.1%) comparisons were in favour of non-participation and 120/141 (85.1%) comparisons were not significant. In addition, 3/37 (8.1%) comparisons found a lower risk of mortality for patients treated inside of RCTs, whereas the remaining 34 (91.9%) comparisons were not statistically significant (Figure 4).

### 3.4. Findings of the Subgroup of Reviews with Low and Critically Low Quality

A review of cancer patients [32] reported 15/27 (55.5%) comparisons in favour of participation and 12/27 (44.5%) comparisons in favour of non-participation. In a general medicine review [28], 10/117 (8.5%) comparisons were in favour of participation, 9/117 (7.7%) were not in favour and 98/117 (83.8%) were statistically non-significant. Mortality was not significant either. In a review [33] focused on the safety of random treatment assignment, 3/25 (12%) comparisons were in favour of participation, 3/25 (12%) were not in favour of participation and 19/25 (76%) were not significant. In addition, in mortality and cancer recurrence, 50% of non-participants died or had a 4-year disease compared to 26% of participants.

## 4. Discussion

Our findings suggest that taking part in RCTs may be beneficial compared to non-participation. This was observed across women’s health, cancer and general medicine RCTs, with evidence from 380 unique comparisons collated in the synthesis. None of the reviews found a harmful effect of participation in RCTs. There was underlying heterogeneity and due to the observational nature of the comparisons, the findings should be interpreted with caution.

To our knowledge, this is the first umbrella review focusing on the benefits of participation in RCTs vs. non-participation. Our search was unrestricted, without limitations regarding the language or time period covered in the databases, to capture the highest possible number of relevant records. There was good reviewer agreement in the search, selection and quality assessments of studies, strengthening the review’s reliability.

In a study by Braunholtz et al. [32], 14 articles reported data from 21 trials, and they concluded that randomised trials tended to have beneficial effects rather than harmful effects on the patients who participated. In addition, a study included in this review showed that survival rates were significantly higher for children within RCTs than for those who were not participating [34]. Similarly, a study comparing survival among cancer patients found better survival in RCT participants compared to patients treated outside of RCTs in the first year after diagnosis [35]. This can be better appreciated in a women’s health meta-analysis [29], in which trial participants compared with non-participants showed improved health outcomes on average. In a high-quality review [30], although in some cases non-participants showed a benefit, a larger number of comparisons reported significantly better outcomes, as well as a lower risk of mortality, in RCT participants. In another high-quality review [33], the same number of studies in favour of participation and in support of non-participation was found; therefore, it cannot be claimed that participants in clinical trials derive a clear and significant benefit. These findings closely resembled those in another review investigating patients with the same disease, treated inside and outside of RCTs [28], in which most of the studies found no statistically significant differences in terms of benefits or harms between participants and non-participants.

The evidence supporting the safety and possible benefits of participation was consistent with the findings of two meta-analyses focused on control group weight changes within lifestyle RCTs. The most recent study showed a slight intragroup weight loss [6]. In a previous study, control groups receiving the usual care lost weight compared to those receiving no intervention, whereas the rest of the control group participants receiving other healthcare protocols did not gain weight [5]. The authors suggested including in future RCTs patient information sheets about the likelihood of weight loss or at least a prevention of weight gain for control group participants [6]. These findings are in alignment with those a previous review [36], in which it was found that most of the comparisons from cancer studies showed an association of trial participation with health benefits, with no evidence of harm. Thus, it has been suggested that the chance of obtaining benefits of participation in clinical trials should be acknowledged to encourage the enrolment of patients in intervention research [37,38]. Patient engagement in healthcare research is likely to be feasible in many settings, although it entails challenges such as the need for increased time and funding [39,40]. Given that randomised trials are necessary in order to provide reliable and high-quality evidence about the effects of clinical interventions [41], it is important to conduct properly designed trials with sufficient sample sizes. It is imperative to inform the eligible population about the benefits or hazards of interventions before their enrolment [42,43,44]. This can be understood as a chance to enhance the health outcomes of participants and to contribute to advances in treatment and healthcare, independently of which participation group they are allocated to. However, this does not imply that all intervention studies had no risk at all, as the hazards and benefits may vary significantly between studies. Understanding why participants exhibited an improved health status would have considerable implications, not only in the interpretation of intervention effects, but also in the design of future intervention trials.

Nevertheless, we acknowledge some methodological limitations. The six selected reviews provided limited evidence, mainly because of the heterogeneity in terms of the quality and size of the comparisons. Clinical variations in the nature of participants, interventions and outcomes can be a strength in terms of generalisability. However, statistical heterogeneity can mask the beneficial effect of trial participation or trial effects. Furthermore, it should be noted that the treatment effect, that is, differences due to interventions received inside instead of outside RCTs, as well as the presence of unmeasured differences in sociodemographic or clinical characteristics between participants in RCTs and non-participants, may affect the interpretation of our findings [45,46]. Given the limited number of reviews available on this topic, and the moderate overlap observed (CCA = 10%), we agreed not to remove any of the included records. This meant that comparisons from systematic reviews shared a 10% of their primary original studies. This proportion represented an acceptable level of redundancy [26].

## 5. Conclusions

Our findings suggest that taking part in RCTs may be beneficial compared to non-participation. Participation in clinical trials should be encouraged and its health impact needs to be addressed in further intervention research. We recommend systematically reporting a comparison between the outcomes amongst participants in RCTs, combining those assigned to control and intervention groups, and those not participating and receiving usual healthcare in a similar setting.

## Figures and Tables

**Figure 1 ijerph-19-15368-f001:**
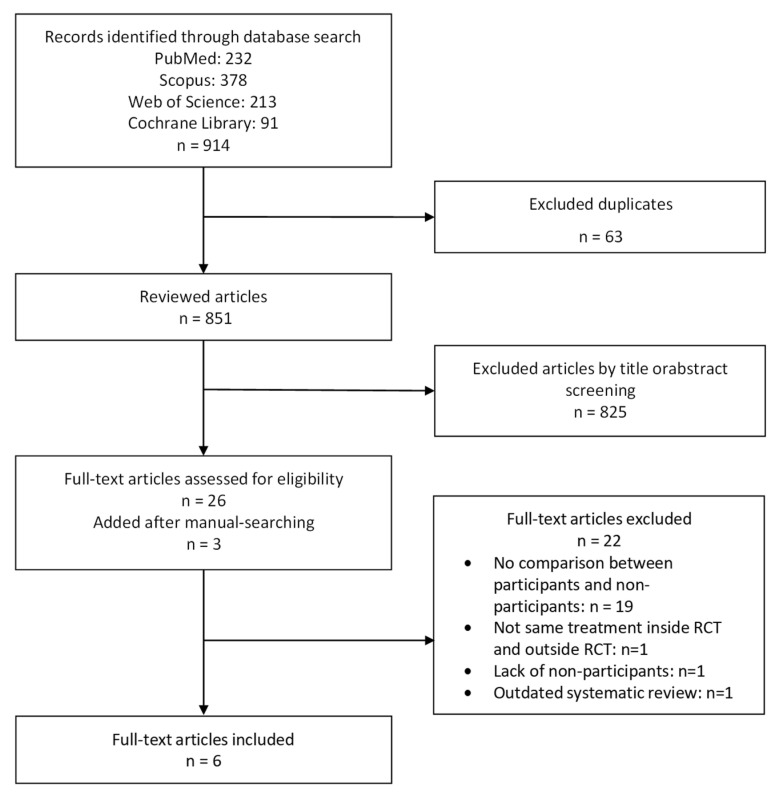
Flow chart of selected reviews and meta-analyses evaluating the benefits of participation in clinical trials.

**Figure 2 ijerph-19-15368-f002:**
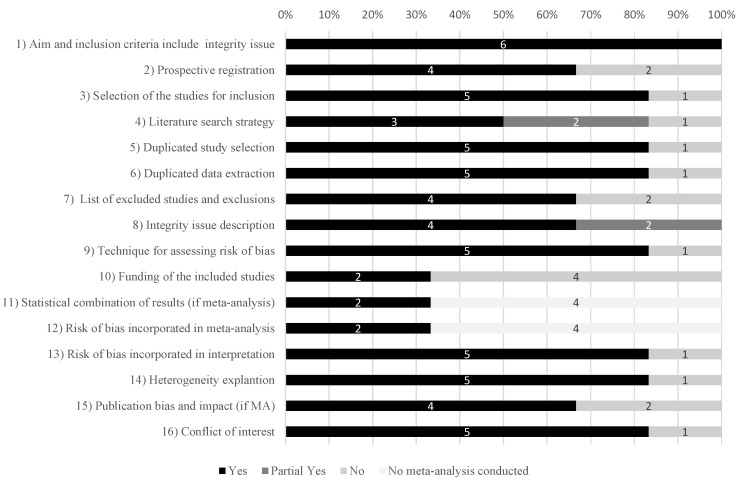
Quality assessment of the selected reviews and meta-analyses evaluating the benefits of participation in clinical trials using AMSTAR-2 (percentage of systematic reviews meeting the16 items).

**Figure 3 ijerph-19-15368-f003:**
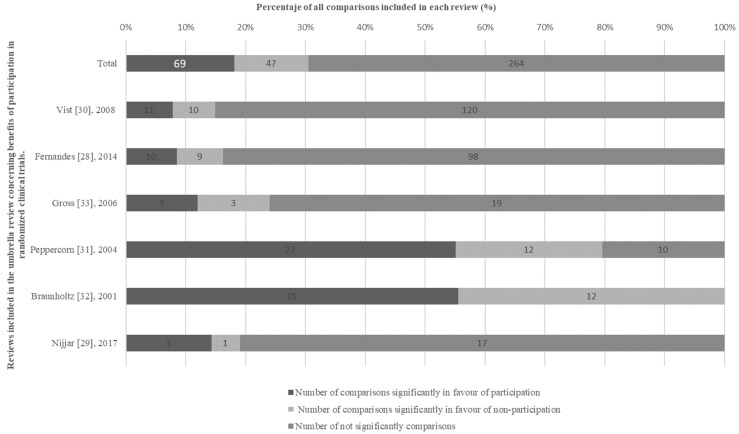
Results of the selected reviews and meta-analyses evaluating the benefits of participation in clinical trials.

**Figure 4 ijerph-19-15368-f004:**
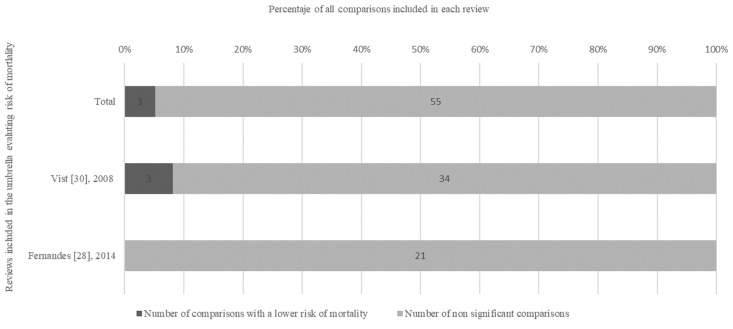
Results of the selected reviews and meta-analysis evaluating risk of mortality.

**Table 1 ijerph-19-15368-t001:** Characteristics of the selected reviews and meta-analyses evaluating the benefits of participation in Randomized Clinical Trials (RCTs).

Author, Year	Objective	No. of Databases Searched, Date Range of Searching, and Publication Data Range (PDR) of Primary Studies	QUALITY ASSESSMent on Primary Studies	RCT Participant Characteristics (No. Comparisons)	Non-Participant Characteristics (No. Comparisons)	Outcomes (Measurement)
Vist [30], 2008	To assess the effects of patient participation in RCTs (’trial effects’) independent both of the effects of the clinical treatments being compared (’treatment effects’) and any differences between patients who participated in RCTs and those who did not	5 databases: The Cochrane Central Register of Controlled Trials (CENTRAL), MEDLINE, EMBASE, The Cochrane Methodology Register, SciSearch and PsycINFO. Up to March 2007. PDR: 1978–2006	Per review. No validated tool used. The criteria followed to assess the validity of comparisons was scored as: “met”, part, “partially met”, “not imbalance”, “not met”, “unclear”.	Patients in different clinical areas and interventions: oncology (31), cardiology (22), other internal medicine subspecialties (27), obstetrics and gynaecology (29), psychology or drug abuse (15), and paediatrics (12), surgery or other procedures (33), drug therapy (28), radiotherapy (15), counselling or education (9), usual care (45), and active monitoring/watchful waiting (6).	Patients receiving similar treatment outside of RCTs (80), eligible refusers (1), patients not invited to participate (2), eligible non-participants who do not meet the above categories (2).	Mortality (dichotomous) morbidity and changes in self-reported pain, quality of life, and function (continuous).
Fernandes [28], 2014	To compare outcomes for patients with the same diagnoses who did (“insiders”) and did not (“outsiders”) enter RCTs, without regard to the specific therapies received for their respective diagnoses.	MEDLINE (1966 to November 2010), Embase (1980 to November 2010), Cochrane Central Register of Controlled Trials (CENTRAL; 1960 to last quarter of 2010) and PsycINFO (1880 to November 2010). From 1880 to 2010. PDR:1979–2009	Per review. No validated tool used.	Patients in different clinical areas and interventions: surgery or medical procedures (46); drug therapy (57); radiotherapy (5); counselling or education (27); other therapy (12).	Patients with the same diagnoses who did not enter RCTs known as “outsiders” (147).	Mortality (dichotomous), patient-reported or other clinically important outcomes (continuous outcomes)
Gross [31], 2006	To quantify the differences in health outcomes between randomized trial participants and eligible non-participants.	Medline, the Web of Science citation database, and manuscript references. From 1984 to 2002. PDR: 1084–2002.	Per review. No validated tool used.	Patients in different clinical areas (oncology, cardiovascular diseases, obstetrics and gynaecology) and interventions: diagnostics (2), medical (14) or surgical (9).	Patients sharing healthcare settings at recruitment, participants recruited in a similar way, eligible non-participants, non-participants allowed to access agents used in a trial.	Mortality, treatment acceptability, and proportion of time or number of days with a given condition.
Peppercorn [32], 2004	To assess the empirical evidence that patients with cancer who enrols in clinical trials have better outcomes than those who do not enrol.	Medline. Search range nor defined. PDR: 1971–2002.	Per review. Not validated tool. Pilot-tested forms recording potential sources of bias.	Cancer patients (24).	Eligible refusers (4), patients in retrospective cohort (21), participants in natural experiment (1).	Health benefits in cancer patients in RCTs (trial effect)
Braunholtz [33], 2001	To assess whether there is evidence that randomized controlled trials are systematically beneficial, or harmful, for patients.	Databases not defined. Up to August 1996. PDR: 1879–1996.	Per review. Not validated tool. Sources of bias of concern were conceptualized as: “patient selection bias”, clinician selection bias”, “detection bias”, “transfer bias” and “study induced bias”.	Patients in cancer therapy (10), cardiovascular (2), other medical interventions (2).	Indirect comparisons (2); patients in at least one concurrent non-trial control group (11): eligible refusers (3), refusers and eligible non-recruited patients (4), all non-randomized patients of recruiting clinicians (1), all non-recruited patients of recruiting and non-recruiting clinicians (3); no control group (1).	Health benefits in cancer patients in RCTs,
Nijjar [29], 2017	To determine whether participation in randomised controlled trials (RCTs), compared with non-participation, has a beneficial effect on women’s health.	MEDLINE (1966 to December 2015), Embase (1980 to December 2015), Cochrane Central Register of Controlled Trials (CENTRAL; 1960 to last quarter of 2015) and PsycINFO (1880 to December 2015). PDR:1981–2015.	Per review. Jadad scale and Newcastle Ottawa score (NOS).	Women in obstetrics-gynaecology interventions: medical (12), surgical (6), and other (3).	Comparable non-participants cohort (21).	Health benefits in women, fetuses or new-borns (dichotomous).

## Data Availability

Not applicable.

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
