# Peer review of "Benefits of Participation in Clinical Trials: An Umbrella Review"

_ijerph, 2022, doi:10.3390/ijerph192215368_

Round 1

Reviewer 1 Report

Thank you for the opportunity to review the manuscript, "Benefits of participation in clinical trials: An umbrella review," submitted to the IJERPH. Although the topic is interesting and important, there are a number of methodological limitations and reporting criteria that need to be addressed in a major revision. In this regard, the specific methodology used for the umbrella review should be discussed as the PROSPERO is a prospective registration. The information provided in the PROSPERO for this umbrella review is limited in terms of the methodology and methods with some seemingly conflicting information as noted in the summary observations.

When applying the PRISMA 2020 checklist for this review, there are many basic reporting elements missing such as the dates for data collection and the PRISMA flow diagram. As there needs to be a more complete reporting of this umbrella review, the authors should complete the PRIOR (Reporting guideline for overviews of reviews of healthcare interventions) checklist. This PRISMA extension for umbrella systematic reviews can be accessed at: https://www.equator-network.org/reporting-guidelines/reporting-guideline-for-overviews-of-reviews-of-healthcare-interventions-development-of-the-prior-statement/. The completed checklist should be returned as a supplemental file with the manuscript. 

SUMMARY OF OBSERVATIONS

Please provide a reference for this statement: Line 44 (Introduction) - Informed consent offered to patients before enrolment into RCTs does not provide information about potential benefits of participation. This process is required to provide an overview of the possible risks (with likelihood) and potential benefits of therapies. As such, the statement seems to be incorrect.

The paragraph from lines 44 to 64 is not focused on a point of significance. Instead, the paragraph provides a list of studies rather than an integration of the literature with evidence to support the point. Please address this observation.

What is the relevance of "patient-centred" care or approach(es) to the topic? the sentences from lines 65 to 68 are not clear about this point. Lines 65-68 (Introduction) - "A patient-centred approach to trial recruitment may help staff build trusting relationships with potential participants and foster mutual commitment. If it can be demonstrated that participating in RCTs improves health status, this would enthuse health professionals to take part in research and be confident about inviting patients to engage in trials 13, 14".

There is not a clearly defined PICO question for the review. The only statement about the purpose statement (or research question) is provided in Line 71 (Introduction) - In this umbrella review we aimed to determine if there was a benefit of participation in RCTs versus non-participation. Yet, in the PROSPERO the PICO question seems to have a different focus than this review - "Does participation in trials improve health outcomes? P: people eligible for trials; I: participation in trials; C: no participation; and O: changes in health outcomes." In this regard, beneficial or harmful was determined by changes in health outcomes. As such, please provide more information about this discrepancy. 

There is not a clearly defined method for conducting this umbrella review (e.g. Joanna Briggs Institute or Cochrane). As the authors realize, the PROSPERO is a prospective registry for systematic review protocols rather than a methodology for reviews. Line 75 (Materials and Methods) - We performed this umbrella review after prospective registration (PROSPERO number: CRD42021287812) and reported it in accordance with relevant guidelines 16,17. In addition, the current manuscript is not reported according to the PRISMA 2020 statement. The manner in which the information presented by Aromataris et al. (2015) was applied to the methodology or methods also needs to be clarified. The PRIOR checklist needs to be applied to this manuscript, and provided as a supplemental file.

The PROSPERO indicates for the Risk of bias (quality) assessment, the AMSTAR 2 and ROBIS 2 scales will be applied. However, this is not the case as stated in line 102 (Materials and Methods) - The quality of included systematic reviews was independently assessed by two reviewers (ABH and NCI) using the 16-item questionnaire: “A Measurement Tool to Assess Systematic Reviews” (AMSTAR-2) 18, 19.

The search dates are not provided.

There is no PRISMA diagram to provide an overview of the screening process by database.

The discussion is largely an extension of the results section.

There is no section addressing limitations. 

According to which source is acceptable? - Line 190: "Given the limited number of reviews on this topic, with a moderate overlapping (CCA=10%), we agreed not to remove any record included. This meant that comparisons from systematic reviews in our umbrella review shared a 10% of primary original research. This proportion represented an acceptable redundancy.

Other Notes

The funding in the disclosures is different than the funding stated in the PROSPERO (the funding from the European Union is not present).

The acknowledgements in the disclosures repeat the funding information which is not an appropriate acknowledgement.

Author Response

Please find the point-by-point response in the file attached.

Reviewer 2 Report

This research would be contributable for the encourgement of participation of RCT,  and shows a lot of effort of researchers. It would be good if author s consider the followings more in detail.

1) When designing clinical trials, both RCT and blinding are considered for the process if available or not.  In this research,  the benefit of participation is investigated by considering only the RCT.  It's recommended to consider the blinding effect with RCT for the benefits of participation. If blinding effect is not considered, it would be better to mention the reason for it.  

2) It's questionable that heterogeneity is analyzed in the item of meta-analysis is not reported in Appendix 3.  It's suggested to be confirmed that only heterogeneity was analyzed even meta-analysis was not.  

3) If it's available, it's suggested to describe the figure such as flow chart or other way for the process of umbrella review.  

Author Response

Please, find the point-by-point response in the file attached.

Round 2

Reviewer 1 Report

Thank you for the opportunity to review the revised version of Benefits of participation in clinical trials: An umbrella review submitted to IJERPH. The first revision process was essential for making a complete assessment of this umbrella review. The review required more time than usual as I needed to review several of the manuscripts excluded from the review as well as those included in the review. However, the additional time was necessary to understand the availability of participant level data in the included reviews and the rationale for excluding several other reviews.

Although many areas of the manuscript have been improved with the recommended revisions and responses to observations from the first review, there continue to be areas requiring refinement for clarity and revisions to provide content to appropriately report an umbrella review. For this reason, I provide some general observations in the next section about important limitations with specific examples and some recommendations. I look forward to reviewing the next version of the manuscript. 

SUMMARY OF REVIEW

The purpose needs to be further refined, "This umbrella review was aimed at assessing if there are health benefits of participation in RCTs, compared to non-participation" (Line 19) to more clearly define health benefits and how harm is or is not applicable. The addition of the PICO question, "In this umbrella review we aimed to determine if among eligible people (population) there was a health benefit (outcome) of participation in RCTs (intervention) versus non-participation (comparison group)" (Line 75) was helpful to make the focus clearer and more specific. However, the role or relationship of harm remains silent in the review. Please consider this point as several reviews excluded from the current review provide data specific to harm for experimental and control / standard of care groups.

Thank you for further clarifying the specific method used for this umbrella review. As I am familiar with the work of Dr. Aromataris due to my work with JBI, several observations need to be addressed in terms of the cited guidance and the methods reported in this review. For this reason, I highly recommend the authors again carefully review the methods and procedures outlined by Aromataris et al. (2015) in comparison to the current review.

As an example of a MAJOR limitation, consider the data abstraction tool for this study in comparison to the guideline from Aromataris et al. (2015). This is a short statement by Aromataris et al. (2015) as an example - "Guided by the data extraction tool, information extracted from each included review should include the following: citation details, objectives of the included review, type of review, participant details, setting and context, number of databases sourced and searched, date range of database searching, publication date range of studies included in the review that inform each outcome of interest, number of studies, types of studies and country of origin of studies included in each review, instrument used to appraise the primary studies and the rating of their quality, outcomes reported that are relevant to the umbrella review question, method of synthesis/analysis employed to synthesize the evidence, and comments or notes the umbrella review authors may have regarding any included study." However, the majority of this information was not abstracted and provided in the materials.

According to the PROSPERO registration, this review has not be started and much of the information continues to conflict with the manuscript. This is a MAJOR limitation which requires correction rather than explanation. For this reason, the solutions to this MAJOR limitation is to remove the statement in Line 79, "We performed this umbrella review after prospective registration (PROSPERO number: CRD42021287812)." or update the PROSPERO information. 

Thank you for providing the PRIOR checklist for this review "We have also completed the Reporting guideline for overviews of reviews of healthcare interventions (PRIOR)21" (Line 80). At this time, there are several reporting elements that need to be addressed with more than "not applicable" statements.

For the risk of bias assessment, please explain how 11b and 11c are not applicable.

11b

Describe the methods used to collect data on (from the systematic reviews) and/or assess the risk of bias of the primary studies included in the systematic reviews. Provide a justification for instances where flawed, incomplete, or missing assessments are identified but not re-assessed.

Not applicable.

11c

Describe the methods used to assess the risk of bias of supplemental primary studies (if included).

Not applicable.

In the synthesis methods, please explain how 12c is not applicable.

12c

Describe any sensitivity analyses conducted to assess the robustness of the synthesized results.

Not applicable

For the reporting bias assessment, please explain how 13 is not applicable.

13

Describe the methods used to collect data on (from the systematic reviews) and/or assess the risk of bias due to missing results in a summary or synthesis (arising from reporting biases at the levels of the systematic reviews, primary studies, and supplemental primary studies, if included).

Not applicable.

Please use the standard PRISMA Flow Diagram (https://prisma-statement.org/prismastatement/flowdiagram.aspx) for Figure 1. In terms of the full text articles excluded, there are 22 indicated to be "irrelevant." This is not the manner in which the PRISMA recommends reporting. There should be more detailed information provided in the diagram aligned with the table. 

15a

Describe the results of the search and selection process, including the number of records screened, assessed for eligibility, and included in the overview of reviews, ideally with a flow diagram.

Line 134-136 (figure 1)

15b

Provide a list of studies that might appear to meet the inclusion criteria, but were excluded, with the main reason for exclusion.

Line 136-138 (Appendix 2)

In terms of the table of excluded studies, the statement "no comparison between participants and non-participants" needs to be further refined. There are at least two studies in the table where the data for participants (experimental group) and the non-participants, which could be no treatment or standard of care groups, are provided.

The synthesis of findings provides no data other than the number of studies finding benefit or harm (the reason for my previous observation about the role of harm). Where is the data about number of participants, median and ranges of numbers, incidents, odds ratios, relative risk, 

"None of the reviews found a harmful effect of participation in RCTs in their overall synthesis. In two of our reviews 31, 32 the majority (>50%) of comparisons were in favour of participation in RCTs. Of the total number of comparisons included, 69 (18.7%) were in favour of participation reporting statistically significant better outcomes for patients treated within RCTs and 264 (71.7%) comparisons were not statistically significant, while 35 (9.5%) comparisons were in favour of non-participation" (Line 151).

The results are weak with only reporting the percentage of reviews rather than outcomes data for benefit and/or harm. This is an especially important limitation when there are only 6 reviews included in the review but 3 reported meta-analyses. Although the appendix 3 provides a list of summarized information for each study, there is not an analysis and synthesis of the data presented as findings. The lack of better quality evidence is a major limitation that needs to be addressed by the authors.

The discussion section needs to be aligned with the findings in terms of bringing other literature into the review. The findings as aligned with the discussion section then need to link to the specific conclusions. As a side note, the limitations area should be presented at the close of the discussion section rather than the current location as the second paragraph.

The conclusion seems to provide a partially accurate statement about harm but there is no specific participant / non-participant level data provided in the findings. As such, the descriptive comparison is not sufficiently strong evidence to make a very specific statement about benefits. In addition, the recommendations are not new in the context of the benefit and harm thresholds established to protect human subjects. The recommendations are not supported by the findings or explicated through the discussion (see the paragraph below).

"Our findings suggest that taking part in RCTs is unlikely to be harmful, and there is a possibility of benefit compared to non-participation. Participation in clinical trials should be encouraged and its health impact need to be addressed in further intervention research. We recommend to systematically report in RCTs a comparison between the outcomes amongst participants, combining those assigned to control and intervention groups, and those not participating and receiving usual healthcare in a similar setting" (Line 245)

Finally, the acknowledgments section is not the appropriate area to provide funding information. This is clearly outlined in multiple guidelines for editors and authors about the dissemination of research results. For this reason, I need to insist the authors comply acknowledge people and organizations in the acknowledgements section without addressing funding. 

Other observations

Line 135 - "Date of searching databases ranged from 1880 28 to 2015 29." Does including a review in the current review with one of the authors for the current review in the review create a possible conflict of interest or bias in the review?

Author Response

Dear reviewer, 

Please, find attached the responses to your valuable comments with the changes made in the revised version of the manuscript, which includes the table 1 and figure 1 modified.

Sincerely,

The authors.
